# COPT: Coordinated Optimal Transport on Graphs

**Yihe Dong**
Microsoft

**Will Sawin**
Department of Mathematics
Columbia University

## Abstract

We introduce COPT, a novel distance metric between graphs defined via an optimization routine, computing a coordinated pair of optimal transport maps simultaneously. This gives an unsupervised way to learn general-purpose graph representation, applicable to both graph sketching and graph comparison. COPT involves simultaneously optimizing dual transport plans, one between the vertices of two graphs, and another between graph signal probability distributions. We show theoretically that our method preserves important global structural information on graphs, in particular spectral information, and analyze connections to existing studies. Empirically, COPT outperforms state of the art methods in graph classification on both synthetic and real datasets.

## 1 Introduction

We introduce a new unsupervised method to measure the distance between a pair of graphs, and apply it to graph sketching. This distance is based on the general notion of optimal transport distance, which involves minimizing a loss function over transport plans between two distributions [21]. However, our distance is defined by minimizing a loss function over *pairs* of simultaneous transport plans, one between the vertices of the two graph and one between distributions defined by the Laplacian spectra of the graphs. This allows us to compare, in a flexible way, large-scale spectral information between the two graphs. Thus, we call it Coordinated OPtimal Transport, or COPT. We show that COPT has desirable properties in theory, as well as empirically demonstrate its usefulness in graph sketching, retrieval, and summarization, on both synthetic and real world datasets.

Constructing a distance metric between graphs and studying its applications come from a long, rich line of work, due to the ubiquity of graph-structured data and the importance of graph sketching and retrieval. We briefly highlight some recent developments in this field, while drawing more detailed connections throughout the text.

Sketching is often defined as choosing a sequence of combinatorial operations (e.g. edge contractions) that minimizes a measure of distance between the sketch and the original graph. For instance, [33] contracts edges in such a way as to preserve the Laplacian spectrum. [7] removes edges and merges vertices in a way that minimizes the Frobenius norm of changes in the psuedoinverse of the Laplacian.

Sketching has been applied to a number of different graph problems. [43] used iterated graph sketching to find optimal orderings of the vertices of a graph. Building on this, [9] defined an efficiently computable notion of distance on graphs. [31] used sketching to efficiently solve linear systems involving the graph Laplacian. Generalizing ideas from electrical engineering, [16] defined a sketch as the Schur complement of the Laplacian with respect to a subset of the nodes. [46] generalized multiscale methods to graphs with extra structure on the nodes. We discuss further connections to related works in §4, as well as provide empirical comparisons to competitive baselines.

Our **main contributions** are 1) devising a coordinated optimal transport algorithm for computing graph distance; 2) applying COPT to graph sketching, obtaining small sketches that allow for improved graph classfication, insightful visualization, and high quality retrieval.

This paper is outlined as follows: In §2, we review generalities on optimal transport methods for graph comparison and discuss prior work. In §3, we define coordinated optimal transport distance and discuss its properties. In §3.3, we describe our approach to graph sketching. In §5, we discuss algorithm implementations and experimental results.

## 2 Graph distances based on optimal transport on vertices

In general, we would like a notion of "distance" for graphs that satisfies the properties of a metric, and in particular is zero if and only if the two graphs are isomorphic. We would also like this distance to be reasonably computable in practice.

Because no simple complete invariant for graphs up to isomorphism is known [2], the most natural approach to define a distance for graphs up to isomorphism is to define the distance between graphs $X$ and $Y$ as a minimum, over bijections between the vertices of $X$ and the vertices of $Y$, of some quantity, which vanishes if and only if this bijection sends the edges of $X$ to the edges of $Y$. For instance, we could take the minimum over bijections of the cardinality of the symmetric difference of the edge set. However, there are some downsides to minimizing over permutations.

First, such a distance would be hard to compute, or even approximate, in practice, as it involves a complicated discrete optimization problem.

Second, such a distance would not be defined if our graphs $X$ and $Y$ have different numbers of vertices.

To solve these problems, we can define a graph distance as a minimization over transport plans. To define these, we first **fix some notation**. Let $X$ be a graph with $N$ vertices and $Y$ a graph with $M$ vertices. We will also use $X$ and $Y$ to denote the set of vertices of $X$ and $Y$ respectively. Optimal transport plans are functions $P$ from $X \times Y$ to $\mathbb{R} \cup \{0\}$ such that $\sum_{x \in X} P(x,y) = N$ for all $y \in Y$, and $\sum_{y \in Y} P(x,y) = M$ for all $x \in X$. We will define distances as a minimum over transport plans $P$, so their formulations will be analogous to the optimal transport, or Wasserstein, distance between the uniform distribution on $X$ and the uniform distribution on $Y$, defined as

$$W_p(X,Y) = \min_{\substack{P:X \times Y \to R^+ \\ \sum_{x \in X} P(x,y)=N \\ \sum_{y \in Y} P(x,y)=M}} \Big( \sum_{x \in X} \sum_{y \in Y} d(x,y)^p P(x,y) \Big)^{1/p} \tag{1}$$

where $d(x,y)$ is a distance function between two points.

**Gromov-Wasserstein distance.** However, before being able to apply (1) to graphs, there is no notion of $d(x,y)$ for $x, y$ vertices in two different graphs.

To fix this, Mémoli proposed a notion of Gromov-Wassestein distance for graphs [37], as

$$\Big( \min_{\substack{P:X \times Y \to R^+ \\ \sum_{x \in X} P(x,y)=N \\ \sum_{y \in Y} P(x,y)=M}} \sum_{\substack{x_1,x_2 \in X \\ y_1,y_2 \in Y}} (d_X(x_1,x_2) - d_Y(y_1,y_2))^p P(x_1,y_1) P(x_2,y_2) \Big)^{1/p}. \tag{2}$$

In other words, given the distance $d_X$ for two vertices in the same graph, defined as the minimum number of edges in a path connecting them, we have a natural notion of distance between two pairs $x_1, x_2$ and $y_2, y_2$ of vertices on the two different graphs as the difference between the distances of the individual vertices.

A generalization of this definition, along with computational methods and applications, was provided by [42], building on computational ideas of [10]. An application to word embeddings was given by [1]. A similar approach, but based more closely on Gromov-Hausdorff distance, was due to Sturm [49].

**Graph Optimal Transport.** The recently proposed GOT [35] graph distance uses optimal transport in a different way. This relies on a probability distribution $\mu^X$, the *graph signal* of $X$ [44, 15], over functions on the vertices of $X$. This distribution is a multivariate Gaussian, with mean zero, whose variance-covariance matrix is a pseudo-inverse $L_X^\dagger$ of the Laplacian $L_X$. They then define, in the case $N = M$, a distance for graphs defined by optimal transport of these probability distributions.

Let $T : \mathbb{R}^X \to \mathbb{R}^Y$ denote a transport plan and $\sigma : X \to Y$ a permutation, [35] defines a distance as

$$W_2(\mu^X, \mu^Y)^2 = \min_{\substack{\sigma:X\to Y \\ \sigma \text{ bijective}}} \inf_{\substack{T:\mathbb{R}^X\to\mathbb{R}^Y \\ T_\#\mu^X=\mu^Y}} \int_{\mathbb{R}^X} \sum_{x\in X}(f(x)-(Tf)(\sigma(x)))^2 d\mu^X(f) \quad (3)$$

Here $f : X \to \mathbb{R}$ denotes an element of the vector space $\mathbb{R}^X$ of functions from $X$ to $\mathbb{R}$, thus $Tf$ is a function from $Y$ to $\mathbb{R}$. (We could work with vectors $v$ instead of functions $x$, where we would write $v_x - (T(v))_y$ instead of $f(x) - (Tf)(y)$. This would require ordering the set $X$ of vertices, and the entries of the vector $v$ would then be the values of the function $f$. The reason we write $f$ as a function rather than a vector is to make clear the invariance of the definition under permutations of $X$ and $Y$, as well as making it easier to express certain constructions.)

## 3   Coordinated optimal transport

Our definition of a new metric on graphs builds on (3), where we replace the permutation $\sigma$ with an optimal transport plan $P$. Thus, our definition involves two different optimal transport plans: $P, T$, hence named *coordinated optimal transport*. We define our distance $\Delta(X, Y)$ by

$$NM\Delta(X,Y)^2 = \min_{\substack{P:X\times Y\to\mathbb{R}^+ \\ \sum_{x\in X}P(x,y)=N \\ \sum_{y\in Y}P(x,y)=M}} \inf_{\substack{T:\mathbb{R}^X\to\mathbb{R}^Y \\ T_\#\mu^X=\mu^Y}} \int_{\mathbb{R}^X} \sum_{x\in X}\sum_{y\in Y}(f(x)-(Tf)(y))^2 P(x,y)d\mu^X(f). \quad (4)$$

Again, we take $\mu^X$ to be a Gaussian with mean zero and variance-covariance matrix $L_X^\dagger$. In the special case that $N = M$ and $P$ is a permutation, this definition reduces to the definition in [35], up to a normalization factor of $\sqrt{N}$. As in [35], this distance is nonconvex. COPT is more general and can be used between graphs of different cardinalities and for sketching.

For *weighted* graphs $X$ and $Y$, we can define COPT exactly the same way, except that the covariance matrix of the graph signal should be the pseudoinverse of the *weighted* Laplacian. This generalization will be important in sketching, even for sketches of unweighted graphs, but we avoid it elsewhere to simplify concepts.

### 3.1   Properties of COPT

We give an analytic formula for computing COPT distance $\Delta(X, Y)$, and show $\Delta(X, Y)$ is a metric. See the supplementary material for full proofs.

**Lemma 3.1.** *Let $X$ and $Y$ be graphs with vertex sets of size $N$ and $M$ respectively. Then*

$$\inf_{\substack{T:\mathbb{R}^X\to\mathbb{R}^Y \\ T_\#\mu^X=\mu^Y}} \int_{\mathbb{R}^X} \sum_{x\in X}\sum_{y\in Y}(f(x)-(Tf)(y))^2 P(x,y)d\mu^X(f)$$
$$= M\operatorname{tr}(L_X^\dagger) + N\operatorname{tr}(L_Y^\dagger) - 2\operatorname{tr}(((L_Y^\dagger)^{1/2}P^T L_X^\dagger P(L_Y^\dagger)^{1/2})^{1/2}) \quad (5)$$

*where $P$ is the matrix with entries $P(x, y)$.*

*Proof summary.* We extend $\mu^X$ and $\mu^Y$ to distributions on the space of functions on $X \times Y$, in such a way that the infimum we are interested in is exactly the Wasserstein distance between these distributions. Because the extended distributions remain multivariate Gaussians, we can use the known Wasserstein distance formula for multivariate Gaussians [50]. By calculating the variance of these extended distributions, and using the cyclic permutation invariance of the traces of powers of a matrix, this reduces to our stated formula. □

Using this analytic formula for the minimum over $T$, we can approximate the coordinated optimal transport distance by using gradient descent to compute the minimum only over $P$.

**Lemma 3.2.** $\Delta(X, Y)$ *is a metric on the set of isomorphism classes of finite graphs.*

*Proof summary.* We check each axiom from the definition of a metric separately. For each of them, our strategy is based on the corresponding step in the proof that the Wasserstein distance is a metric. Whatever construction must be applied to the transport map or joint measure in the Wasserstein distance proof is applied to both $P$ and $T$ in our proof. For instance, to check the triangle inequality, we compose the transport maps $T$ and also compose $P$ by a matrix multiplication: Given $P : X \times Y \to \mathbb{R}$ and $Q : Y \times Z \to \mathbb{R}$, we take $\frac{1}{M} \sum_{y \in Y} P(x,y)Q(y,z) : X \times Z \to \mathbb{R}$. The calculations in each step are similar to, but more intricate than, the calculations in the Wasserstein distance proof. $\qquad\square$

### 3.2 Global information: spectral vs. metric

We analyze the graph global structures that COPT preserves, and compare and contrast COPT with the state of the art Gromov-Wasserstein (GW) distance [51, 37, 42]. The COPT metric and GW metric are both optimal transport metrics for graphs. The main difference is in what information about the graph they emphasize. The GW distance is defined in terms of the metric $d_X$, and so it measures primarily changes to the graph that change the distance function by a large amount, while COPT is defined in terms of $L_X^\dagger$, so it measures primarily changes to the graph that change the eigenvectors of the Laplacian with *small* eigenvalue by a large amount.

To see the difference between these two concepts, consider a graph with two clusters. The distance between a point in the first cluster and a point in the second cluster is determined mainly by the *length of the shortest path* between the clusters. Adding new paths between the clusters will not change the distance much, while lengthening all paths will change the distance drastically. On the other hand, the entries of the matrix $L_X^\dagger$ with row in the first cluster and column between the cluster is determined more by the *number of paths* between these clusters. As we add more and more paths, these entries of $L_X^\dagger$ will get less and less negative, up until the number of paths between the clusters is almost as large as the number of paths within a cluster. However, lengthening the paths will affect $L_X^\dagger$ less. [1]

To see the relationship between the graph Laplacian and counting short paths, it is convenient to use the following geometric series expansion:

$$L_X^\dagger = (D_X - A_X)^{-1} = D_X^{-1} + D_X^{-1} A_X D_X^{-1} + D_X^{-1} A_X D_X^{-1} A_X D_X^{-1} + \dots$$

where $A_X$ is the adjacency matrix and $D_X$ is a diagonal matrix whose diagonal entries are the degrees of each vertex. Thus, each entry of the Laplacian pseudoinverse is a formal series counting paths, where for instance the entries of $D_X^{-1} A_X D_X^{-1} A_X D_X^{-1}$ are a weighted count of paths of length two. (As long as $X$ is connected and not bipartite, this sum converges once we orthogonally project each term onto the complement of the all 1s matrix, as doing this removes the influence of the all 1s eigenvector, and then we can use the convergence of the geometric series.)

### 3.3 COPT for graph sketching

**Motivation.** Graph sketching replaces a graph with a structurally similar graph with a smaller number of vertices. Many sketching methods focus on preserving the spectrum of the graph, but the best similarity metric may depend on the task. Graph sketching techniques have wide applications, beyond what is discussed in §1, it has also been used to reduce computational load and memory footprint [25], as part of graph convolution networks to learn a hierarchical scaling of graph representations and reduce overfitting [47, 8, 12, 11], and as a key subroutine in graph partitioning [18, 22, 27, 14].

Using the COPT distance function between graphs, we define a method to sketch a graph by reducing it to a low-dimensional matrix, i.e. the sketched Laplacian. The sketched Laplacian preserves key spectral information about the graph. Given a graph $X$ on $N$ vertices and a target size $M$, we search for the graph $Y$ on $M$ vertices that minimizes our distance function $\Delta(X,Y)$. In theory, this graph would be the $M$-vertex graph that best approximates $X$, and therefore should share many of the same features (e.g. clusters or the lack thereof), but with fewer vertices.

**Algorithm 1** COPT graph sketching and graph distance

---

**Input:** Graph $X$ of size $N$, target sketch dimension $M$

**Initialize:** $L_X^\dagger \leftarrow$ inverse Laplacian of $X$

**Initialize:** $(L_Y)'$: the $M(M-1)/2$ strict upper triangular entries of $L_Y$, drawn from $\mathcal{N}(0,1)$

**Initialize:** $P(x,y)$ for $x \in X, y \in Y$, sampled from Uniform$(1,2)$

**for** $i = 1$ **to** n_iter **do**

    **Set** $P(x,y) = \text{abs}(P(x,y))$

    **Normalize** $P(x,y)$ by 5 iterations of Sinkhorn-Knopp algorithm

    **Ensure Laplacian properties:** for $y_1 < y_2$, $(L_Y)_{y_2 y_1} \leftarrow -(L'_Y)^2$, $(L_Y)_{y_1 y_2}, \leftarrow (L_Y)_{y_2 y_1}$, $(L_Y)_{y_1 y_1}, \leftarrow -\sum_{y_2 \neq y_1} L_{y_1 y_2}$

    **Minimize** COPT distance $\Delta(X,Y)$ in Eq (4):

    **I:** Compute gradient of Eq (5) evaluated at $L_Y$ and $P(x,y)$

    **II:** Update $L'_Y$ and $P(x,y)$ using gradient

**Return:** $L_Y$: Laplacian of sketched graph; $\Delta(X,Y)$: distance between graphs $X$ and $Y$; $P(x,y)$: the transport plan

---

**Method.** However, there are two problems with reducing to a smaller graph: 1) this is a discrete optimization problem, and continuous optimization problems are often computationally simpler; 2) the number of isomorphism classes of graphs on a small vertex set is relatively small, so smaller graphs cannot preserve much information.

Both of these problems disappear if we reduce to a smaller *weighted* graph - choosing the weights to minimize the distance is a continuous optimization problem, and the weights can provide more information than simply whether an edge exists or not. Since our distance function depends only on the Laplacian $L_Y$ of $Y$, it is convenient to describe this optimization in terms of the Laplacian $L_Y$ that minimizes the distance. We choose $L_Y$ subject to the conditions typical of the Laplacian matrix of a (weighted) graph - it is symmetric, its off-diagonal entries are nonpositive, and its row and column sums vanish.

Formally, the sketch of the graph $X$ is given by the $L_Y$ which attains the minimum

$$
\min_{\substack{L_Y \in M_Y(\mathbb{R}) \\ (L_Y)_{y_1 y_2} = (L_Y)_{y_2 y_1} \\ (L_Y)_{y_1 y_2} \leq 0 \text{ if } y_1 \neq y_2 \\ \sum_{y_2} L_{y_1 y_2} = 0}} \min_{\substack{P: X \times Y \to R^+ \\ \sum_{x \in X} P(x,y) = N \\ \sum_{y \in Y} P(x,y) = M}} \left( M \operatorname{tr}(L_X^\dagger) + N \operatorname{tr}(L_Y^\dagger) - 2 \operatorname{tr}(((L_Y^\dagger)^{1/2} P^T L_X^\dagger P (L_Y^\dagger)^{1/2})^{1/2}) \right)
$$

(6)

In practice, we use a gradient descent algorithm on both $P$ and $L_Y$ simultaneously to find an approximate minimum.

Note that this use of sketching refers to reducing the number of *vertices* in a graph, as opposed to reducing the number of edges as in Laplacian sketching [4, 20], where one finds graph sparsifiers by building a spectrally similar graph with the same number of vertices but a reduced number of edges.

### 3.4 Implementation

Algorithm 1 describes the COPT routine for both computing graph distance and finding the optimal sketch - the only difference being that, when computing the distance between two graphs $X$ and $Y$, $L'_Y$ is given as input and not updated in step II.

As outlined in Algorithm 1, the values of the transport plan $P$ are initially uniformly sampled from the interval $[1, 2]$. At the beginning of each iteration, we normalize $P$ so its row and column sums are equal, by using the Sinkhorn-Knopp algorithm [10]. This ensures that $P$ is a transport plan.

$L'_Y$ corresponds to the upper triangular part of $L_Y$, as that entirely determines the Laplacian. $L'_Y$ is initialized from the standard Gaussian. At the start of each iteration, $L_Y$ is obtained by taking its upper triangular part to be $-(L'_Y)^2$, then symmetrized, and diagonal terms filled, to ensure it's a Laplacian matrix.

Gradient descent is used to minimize the analytic formulation Eq (5), where $L'_Y$ and $P$ are updated at each step, with the Adam optimizer [23] with a multistep learning rate scheduler that reduces

the learning rate multiplicatively at regular intervals. When using COPT to determine the distance between two graphs with Laplacians $L_X$ and $L_Y$, only $P$ is optimized over in each iteration. Our implementation uses PyTorch and one P100 GPU, on a 2.60GHz six-core Intel CPU machine.

As sanity checks, we confirm that 1) the sketched Laplacian $L_Y$ converges to the original graph's Laplacian $L_X$ when the target sketch dimension is that of the original graph, and 2) the distance converges to 0 when $L_X$ and $L_Y$ are fixed to be equal. The effects of $P$ as a transport plan can be seen from the node labels in sketched graphs in Figure 4.

### 3.5   Time complexity

We estimate the coordinated optimal transport distance by a gradient descent algorithm. The time complexity is given by (number of iterations $\times$ time to calculate each iteration). We are unaware of a general method to estimate the number of iterations needed to converge (in practice $\sim$150 iterations suffice to sketch 50-node to 15-node graphs, and $\sim$1000 iterations to sketch 1000-node to 200-node graphs), so we focus on estimating the time per iteration, where the bottleneck is evaluating

$$M \operatorname{tr}(L_X^\dagger) + N \operatorname{tr}(L_Y^\dagger) - 2 \operatorname{tr}(((L_Y^\dagger)^{1/2} P^T L_X^\dagger P (L_Y^\dagger)^{1/2})^{1/2})$$

and its derivative with respect to $P$. This can be done in matrix multiplication time $O(\max(N, M)^\omega) \leq O(\max(N, M)^{2.373})$ [29]. To see this, note that computing the inverse of a matrix can be done in matrix multiplication time, and that these pseudoinverses can be computed by orthogonally projecting onto the complement of the all 1s vector and then taking a usual inverse. Furthermore, the trace of the square root of a matrix is the sum of the square roots of the eigenvalues, and the eigenvalues can be computed in matrix multiplication time [40], noting that we do not need the most computationally difficult step (c) of [40], which computes the eigenvectors. Using back-propagation, computing the gradient has the same time complexity as computing the function. Note quantities such as $L_X^\dagger$ in the above equation can be cached as they do not change across iterations.

## 4   Connections to prior work

The closest analogue of COPT is [36], which also builds on GOT. One difference is that COPT ensures that the mass of the larger graph is evenly distributed over the smaller graph, while [36] allow different vertices to carry different amounts of mass. [36] also does not use the resulting distance for sketching, like COPT does.

It is important to distinguish between graph distances defined using convex optimization, such as those defined in [5], and nonconvex optimization, such as COPT. Both can be relaxations of optimization problems over permutations. For one natural loss function, [34] showed that a nonconvex relaxation better approximates the optimum permutation than a convex relaxation - in fact, with high probability, the convex relaxation is not a permutation at all. While the loss function in [34] is somewhat different from COPT's, we expect the same distinction between convex and nonconvex optimization to apply in our case.

However, the specific distance studied by [34] is very different from COPT. That distance, which was also used by [54], is formally similar to GW, but it uses the adjacency matrices directly instead of the graph metrics $d_X, d_Y$. Thus, the distance between a graph $G$ and $G$ together with one additional edge $e$ will be similar regardless of the location of $e$, while in COPT and GW it will be larger if $e$ connects two clusters that were far apart in $G$. This makes it analogous to the classical graph edit distance, where the cost to add an edge is independent of the location of the edge, and is defined using discrete permutations instead of continuous transport maps.

The graph signal, as used in COPT, can be compared to a graph embedding. In fact, the graph signal defines an embedding of the set of vertices into a vector space of random variables. This space has coordinates given by the eigenvectors of the graph Laplacian, with the distance along the $i$ coordinate weighted by $\lambda_i^{-1/2}$ (where $\lambda_i$ is the $i$th eigenvalue). This can be compared to the graph embedding based on the first $k$ eigenvectors of the Laplacian, which is equivalent to weighting the first $k$ coordinates by 1 and all other coordinates by 0. [39] used this embedding to define a graph distance. Unlike COPT, this distance is not invariant under changes of coordinates and thus is discontinuous as a function of the adjacency matrix whenever an eigenvalue has multiplicity $> 1$.

The first $k$ eigenvalues of the Laplacian were used to measure similarities between graphs by both [30] and [32], although they were not viewed as coordinates of a graph embedding in those papers. These papers construct a sketch by searching for a smaller graph that optimizes their spectral similarity measures, subject to combinatorial restrictions on the structure of the graph sketch. COPT performs a similar search but without combinatorial restrictions.

Along similar lines but even further from COPT is [33], which defines a sketch as a purely combinatorial process where randomly chosen edges are contracted. Still, [33] proves upper bounds on the difference between the spectra of the original graph and the sketch with high probability.

Graph embeddings that are not necessarily based on the eigenfunctions of the Laplacian have also been used to compare graphs, such as in the work of [55], which combines arbitrary graph embeddings with Gromov-Wasserstein distance.

Sketching based on optimal transport was also used in [17], which chooses the sketch to be a subgraph of the original graph, unlike COPT where the sketch is a new graph. Because of this, it can use the usual Wasserstein optimal transport distance as a distant function. Like the GW distance, this approach preserves largely metric, rather than spectral, information on graphs.

## 5 Experiments

COPT can be used both for sketching when given one graph, and finding the distance when given two graphs. Here we demonstrate its effectiveness on a variety of tasks: sketching, retrieval, classification, and summarization. Additional experiments, such as using low-dimensional COPT sketches to visualize relations between graphs, and an extensive comparison with GOT [35], can be found in the supplementary material.

### 5.1 Graph Sketching

We measure COPT sketching quality and compare with state of the art techniques: OTC [17], an OT-based compression method that uses Boolean relaxations to create a compressed graph that's a *subgraph* of the original graph; variation neighborhood (Variation) [32], a combinatorial optimization approach to graph coarsening; REC [33], a randomized edge contraction algorithm that preserves the spectrum; algebraic distance (Algebraic) [9, 43], which contracts edges based on weights calculated using the Jacobi method; affinity (Affinity) [31], a vertex proximity heuristic; and heavy edge matching (HeavyE) [13], an edge contraction algorithm based on the weight of an edge and the degrees of its joining vertices.

We determine the sketching quality by measuring the graph classification accuracy on sketched graphs. Specifically, for each of the benchmark algorithms and each dataset, we first sketch the graphs by a given compression factor, then use 70% of the sketched graphs to train an SVM with the multiscale Laplacian graph kernel [24, 26, 52], a kernel able to incorporate structural information of neighborhoods in the graph over a range of sizes. Finally we test the classification accuracy on the remaining 30% of sketched graphs. This is done for both 2- and 4-fold compression.

This is done on four benchmark datasets over diverse domains: Proteins [6], BZR_MD [26], MSRC_9 [38], and Enzymes [45]. The SVM is trained with parameters found using 3-fold cross validation on the training set, using a fast approximation of the multiscale Laplacian kernel (using the Nyström Method [53]). Each accuracy measurement is repeated five times. As shown in Table 1, COPT performs competitively across datasets for both 2- and 4-fold vertex reduction. In particular, the fact that COPT performs strongly in the 4X compression case could be due to the fact that COPT achieves a continuous, not discrete, relaxation with a weighted Laplacian.

### 5.2 Fast Graph Retrieval

It is often useful to reduce a set of graphs to the *same number of vertices* rather than by the same compression factor (reducing to different numbers of vertices), such as for fast similarity measure between graphs using a simple $l^1$ or $l^2$ distance; or for neural network training, where batched operations are the norm and common operations such as MLP require the input dimensions be the same across samples. Uniformized dense data can also be processed more efficiently on GPUs than sparse data [56].

| | 2X compression | | | | 4X compression | | | |
|---|---|---|---|---|---|---|---|---|
| | BZR_MD | MSRC_9 | Proteins | Enzymes | BZR_MD | MSRC_9 | Proteins | Enzymes |
| OTC | 60.7±4.0 | 80.9±4.5 | 72.8±.8 | 29.1±4.6 | 64.3±2.7 | **84.8±6.7** | 66.7±1.8 | 25.2±2.7 |
| HeavyE | 61.7±4.8 | 79.7±6.3 | 72.3±3.3 | 27.8±2.3 | 55.0±4.7 | 76.1±7.9 | 72.2±2.7 | 24.9± 1.9 |
| Variation | 60.2±4.4 | 75.5±2.7 | 72.1±1.2 | 31.7±1.5 | 59.3±3.2 | 78.5±3.8 | 72.4±.75 | 27.7±2.1 |
| Algebraic | 57.4±5.2 | 77.0±8.9 | 70.1±2.7 | **35.1±2.3** | 53.4±2.5 | 75.2±6.9 | 69.1±1.8 | 24.3±2.7 |
| Affinity | 58.5±5.0 | 80.1±3.0 | 71.2±2.5 | 25.2±1.5 | 53.4±3.5 | 75.8±6.2 | 70.9±2.3 | 23.5±4.0 |
| REC | 60.9±7.3 | 82.4±1.9 | 71.1±1.5 | **34.7±2.4** | 54.5±2.7 | 77.9±3.7 | 71.5±1.0 | 28.9±1.8 |
| COPT | **67.6±4.0** | **86.3±1.3** | **74.0±1.3** | 32.2±3.3 | **68.4±5.0** | 81.2±4.8 | **73.7±1.5** | **33.1±4.2** |

Table 1: Graph classification accuracy (mean ± standard deviation) comparison with state of the art techniques on datasets across diverse domains, when the number of vertices is reduced by 2- and 4-fold. Accuracies reported in %.

**Synthetic dataset.** We test the quality of COPT sketching to *equal* number of vertices by graph retrieval quality, as judged by accuracy of the class of the nearest neighbor. Specifically, we take the dataset $\{G_X\}$ and queries $\{G_Q\}$ to be 600 and 180 randomly generated 50-node graphs, respectively, each evenly distributed amongst six classes: random geometric [41], block-2 , block-3, block-4 [19], Barabasi-Albert [3], and random regular graphs [48]. We vectorize each graph $G$ in two ways: 1) sketch $G$ to 15 nodes with COPT, and flattening the upper triangular part of the sketched Laplacian to obtain a 120-dimensional vector, 2) take the spectral projection of $G$'s Laplacian, specifically the eigenvectors corresponding to the three smallest non-zero eigenvalues (the zero eigenvalue corresponds to the constant eigenvector), yielding a 150-dimensional vector. Smallest eigenvalues are taken as the lower spectrum corresponds to global structure. Given a query graph vector $v_q$, we take its predicted class to be the class of its nearest neighbor, where the distance is determined with $l^1$ distance for COPT sketches, and $l^2$ for spectral projections.

**Results.** When taking the nearest neighbor's class as the predicted class, COPT sketching achieves $97.8 \pm 1.1\%$ accuracy, which is $15.7\%$ higher on average than the spectral projections accuracy of $82.8 \pm .6\%$. This is repeated three times. To illustrate the speed advantage of equi-dimensional sketching, note that retrieval accuracy on COPT sketches trails only $2.0\%$ behind the GW accuracy of $99.8 \pm .3\%$ on the *original, non-reduced* graphs, while being 2000X faster: $1.81 \pm .07$ ms compared to $3.69 \pm .07$ s. Thus COPT is ideally suited for situations where fast execution speed is critical, e.g. as a component in a pipeline.

**Combining in pipeline.** In practice, a faster, but coarser, algorithm is often used to filter out candidates for a more accurate but time-consuming method [28], so we report accuracies for pipelines that 1) does either $l^1$ retrieval on COPT sketches or $l^2$ retrieval on spectral projections to filter out unlikely candidates, and 2) runs GW on the remaining candidates. Retrieval using COPT sketched Laplacians significantly outperforms spectral projections of original Laplacians, For instance, when the fast algorithm is allowed to filter out all but the top 3 candidates, COPT+GW pipeline achieves $98.7 \pm 0.3\%$ accuracy, compared with $89.4 \pm 2.4\%$ for spectral projections+GW, with comparable timings as the compute bottleneck lies in the GW component of the pipeline. See appendix for the full comparison.

**Real dataset.** We also compare reduction to equal dimensions on the real dataset BZR_MD, with on average 21.3 nodes per graph. When reduced to 7 nodes per graph and the upper triangular part of the reduced Laplacian taken as above, COPT achieves $57.2 \pm 4.9\%$ accuracy in the class of the nearest neighbor, compared to $52.2 \pm 5.5\%$ for OTC [17]. Here 100 of the 306 graphs in the dataset are sampled to query the remaining 206 graphs. This is repeated 20 times.

## 5.3 Graph Summarization

Figure 4 visually demonstrate that COPT preserves the most relevant global structures on graphs, across graphs of varying global structures. The sketched graph visualizations are obtained from Algorithm 1 by declaring an entry in the sketched Laplacian $L_Y$ an edge if it lies above a given threshold, where the threshold is determined based on a gap in the values distribution of $L_Y$. The node labels on the sketched graphs are determined using the transport plan $P$, specifically the label on a node contains the two top-weighted nodes in the original graph whose mass flowed into that node. This shows that 1) COPT preserves important global graph structures, and 2) structurally similar

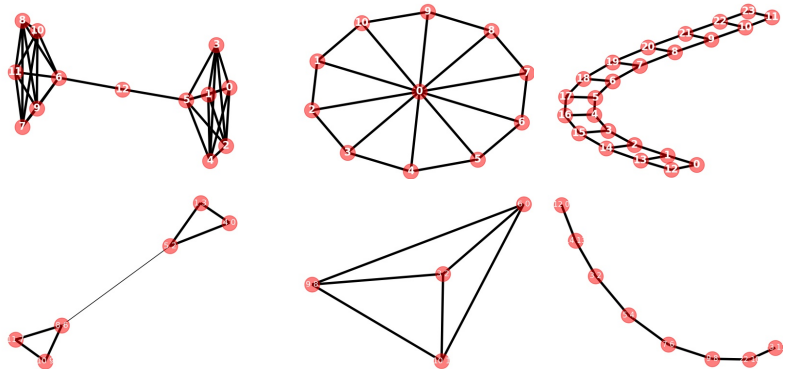

Figure 1: Barbell.          Figure 2: Wheel.          Figure 3: Ladder.

Figure 4: Orginal graphs (top) and their sketched graphs. The node labels on the sketched graphs are determined using the transport plan $P$, specifically the label on a node contains the two top-weighted nodes in the original graph whose mass flowed into that node. COPT sketches structurally similar nodes in the original graph to the same or nearby nodes.

nodes in the original graph are sketched to the same or nearby nodes. See supplementary material for more examples.

# 6    Conclusion

COPT is a novel framework for creating new graphs based on existing graphs and drawing relations amongst graphs, leveraging key graph structural information. This work opens the door to a variety of exciting future directions. For instance, in additional to the graph spectral information, additional information that often materializes in real world data can be leveraged, such as *node attribute* information. Currently, a naive way to incorporate node attributes is to use the transport plan $P$, for *sketching* this means deriving the attribute of a node in the sketch from the attributes of the nodes in the original graph whose mass flowed to that node in the sketch. For graph *comparison* this means comparing the similarity of attributes of nodes in the origin and target graphs that are matched by $P$, taking this attribute similarity into account per iteration when optimizing $P$. There are perhaps more organic ways of doing this, for instance one can conceive of ways to fuse COPT with another construct, such as the Gromov-Wasserstein distance. Another interesting and important direction is to scale up COPT, for instance computing COPT iterations for a batch of graphs simultaneously, which is nontrivial currently, as the transport map joint optimization is specific for a given pair of graphs.

## Acknowledgments and Disclosure of Funding

We are very grateful for the anonymous reviewers for suggestions that improved the clarity of the paper, including changes both in technical exposition and paper organization. This research was conducted during the period Will Sawin served as a Clay Research Fellow.

## Broader Impact

Graph-structured data are ubiquitous, thus fast and accurate graph retrieval and comparison is an important application. COPT improves upon state of the art methods on real world datasets such as Proteins and Enzymes, this can be useful for both scientific research and medical applications, such as comparing a novel synthesized protein with existing ones or trying to identify a molecule. As the COPT metric can be used to compare graphs with different numbers of vertices and is invariant under permutations of the vertices, it can be applied to a broad spectrum of graphs. In addition, the fact that equidimensional COPT sketches achieve competitive retrieval accuracy at a fraction of the time compared to state of the art methods makes it suitable for retrieval in large-scale datasets. However,

caution is warranted to avoid over-interpreting COPT or other graph distances - if the distance is low, that only implies structural similitary, not necessarily semantic similarity.

## Footnotes

[1]Consequently, for downstream applications, metric approximations are perhaps better suited when shortest paths between nodes are important, e.g. classifying road networks, classifying graphs arising from physical objects where edge lengths carry geometric information. Spectral approximations are likely more useful when the number of paths matter more, for instance graph clustering, or graph partitioning, where one aims to minimize the number of edges cut (one way is to iteratively coarsen graphs and find cuts on smaller graphs).

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
