[Supplementary Material]

# COPT: Coordinated Optimal Transport on Graphs
# Supplementary Material

## Supplement Outline

In this supplement, we give full proofs of Lemmas 3.1 and 3.2, further discuss COPT optimizations during training, as well as give additional results on graph summarization and visualiation, training progress, and comparison between COPT and GOT, including both theoretical and empirical results that GOT is a special case of COPT when $N = M$.

## 1 Full proofs to lemmas

We give full proofs to Lemma 3.1, an analytic formula for the COPT metric, and Lemma 3.2, the statement that COPT is a metric.

**Lemma 1.1** (Lemma 3.1). *Let $X$ and $Y$ be graphs with vertices sets of size $N$ and $M$ respectively. Then*

$$\inf_{\substack{T:\mathbb{R}^X \to \mathbb{R}^Y \\ T_{\#}\mu^X = \mu^Y}} \int_{\mathbb{R}^X} \sum_{x \in X} \sum_{y \in Y} (f(x) - (Tf)(y))^2 P(x,y) d\mu^X(f)$$

$$= M \operatorname{tr}(L_X^{\dagger}) + N \operatorname{tr}(L_Y^{\dagger}) - 2 \operatorname{tr}(((L_Y^{\dagger})^{1/2} P^T L_X^{\dagger} P (L_Y^{\dagger})^{1/2})^{1/2})$$

*where $P$ is the matrix with entries $P(x,y)$.*

*Proof.* Let $A$ be the map from $\mathbb{R}^X$ to $\mathbb{R}^{X \times Y}$ that sends a function $f$ on $X$ to the function $f(x)\sqrt{P(x,y)}$ on $X \times Y$. Similarly, let $B$ be the map from $\mathbb{R}^Y$ to $\mathbb{R}^{X \times Y}$ that sends a function $g$ to $g(y)\sqrt{P(x,y)}$. Then the distance between $A(f)$ and $B(Tf)$ in $\mathbb{R}^{X \times Y}$ is

$$\sum_{x \in X} \sum_{y \in Y} (f(x)\sqrt{P(x,y)} - (Tf)(y)\sqrt{P(x,y)})^2 = \sum_{x \in X} \sum_{y \in Y} (f(x) - (Tf)(y))^2 P(x,y).$$

Thus, we can interpret this minimum as the Wasserstein distance between the pushforward $A_{\#}\mu^X$ of $\mu^X$ along $A$ and the pushforward $B_{\#}\mu^Y$ of $\mu^Y$ along $B$. Because $A$ and $B$ are linear maps, these distributions are both multivariate Gaussians with mean zero. Thus, we can use the formula for the Wasserstein distance between multivariate Gaussians with mean zero, which is expressed in terms of their covariance matrices [8, Theorem 2.2 and Remark 4.2]. The formula is

$$\operatorname{tr}(V(A_{\#}\mu^X)) + \operatorname{tr}(V(B_{\#}\mu^Y)) - 2 \operatorname{tr}\left(\left(V(A_{\#}\mu^X)^{1/2} V(B_{\#}\mu^Y) V(A_{\#}\mu^X)^{1/2}\right)^{1/2}\right).$$

We have

$$V(A_{\#}\mu^X) = A L_X^{\dagger} A^T$$

and

$$V(B_{\#}\mu^Y) = B L_Y^{\dagger} B^T$$

and we have by direct calculation

$$A^T A = M I_N,$$

$$B^T B = N I_M,$$
$$B^T A = P,$$
$$A^T B = P^T,$$

giving

$$\operatorname{tr}(V(A_{\#}\mu^X)) = \operatorname{tr}(AL_X^\dagger A^T) = \operatorname{tr}(L_X^\dagger A^T A) = M \operatorname{tr}(L_X^\dagger)$$
$$\operatorname{tr}(V(B_{\#}\mu^Y)) = \operatorname{tr}(BL_Y^\dagger B^T) = \operatorname{tr}(L_Y^\dagger B^T B) = N \operatorname{tr}(L_Y^\dagger)$$

For the last term, it is convenient to define the trace of the square root of a matrix as the sum of the square roots of its eigenvalues, so that it can be defined for more than just symmetric positive definite matrices, allowing us to write

$$\operatorname{tr}\left( \left( V(A_{\#}\mu^X)^{1/2} V(B_{\#}\mu^Y) V(A_{\#}\mu^X)^{1/2} \right)^{1/2} \right) = \operatorname{tr}\left( \left( V(A_{\#}\mu^X) V(B_{\#}\mu^Y) V \right)^{1/2} \right)$$

$$= \operatorname{tr}\left( \left( AL_X^\dagger A^T B L_Y^\dagger B^T \right)^{1/2} \right) = \operatorname{tr}\left( \left( B^T A L_X^\dagger A^T B L_Y^\dagger \right)^{1/2} \right)$$

$$= \operatorname{tr}\left( \left( P L_X^\dagger P^T L_Y^\dagger \right)^{1/2} \right) = \operatorname{tr}\left( \left( \left( L_Y^\dagger \right)^{1/2} P L_X^\dagger P^T \left( L_Y^\dagger \right)^{1/2} \right)^{1/2} \right).$$

Combining these, we get exactly the stated formula. Note that, in our final formula, the matrix is again semidefinite, so we can take a canonical square root. □

**Lemma 1.2** (Lemma 3.2). *$\Delta(X, Y)$ is a metric on the set of isomorphism classes of finite graphs.*

*Proof.* If $X = Y$ then the distance is 0, because then $L_X = L_Y$ and we can take $P$ to be the diagonal matrix $N I_N$.

Conversely, let us check that if the distance is zero then $X = Y$. Because the distance is the minimum of a continuous function over the compact set of possible values of $P$, if the distance is zero then zero is obtained for a particular value of $P$. Because the Wasserstein distance between two distributions is zero if and only if they are the same distribution, it follows that (in the notation of the previous lemma) $AL_X^\dagger A^T = BL_Y^\dagger B^T$, which in concrete terms means that

$$\left( L_X^\dagger \right)_{x_1 x_2} = \left( L_Y^\dagger \right)_{y_1 y_2}$$

whenever $P(x_1, y_1) \neq 0$ and $P(x_2, y_2) \neq 0$. Taking $y_1 = y_2 = y$, this implies

$$\left( L_X^\dagger \right)_{x_1 x_2} = \left( L_X^\dagger \right)_{x_1 x_1}$$

whenever $P(x_1, y) \neq 0$ and $P(x_2, y) \neq 0$. This identity implies $x_1 = x_2$, because the unique maximum of a column of $L_X^\dagger$ is on the diagonal. Thus, we see that $P(x, y) \neq 0$ for a unique $x$ for each $y$. By symmetry this must also be true for a unique $y$ for each $x$. Because $P(x, y) \neq 0$ for at least one $y$ for each $x$, and at least one $x$ for each $y$, the set where $P(x, y) \neq 0$ defines a permutation. After applying that permutation, our identity can be written

$$L_X^\dagger = L_Y^\dagger$$

which implies

$$L_X = L_Y$$

and thus

$$X = Y.$$

Symmetry is easiest to check using Lemma 3.1 and the fact that the trace of the square root of a matrix, like the trace of a matrix, is invariant under cyclic permutations, so

$$\operatorname{tr}(((L_Y^\dagger)^{1/2} P^T L_X^\dagger P (L_Y^\dagger)^{1/2})^{1/2}) = \operatorname{tr}(((L_Y^\dagger)^{1/2} P^T (L_X^\dagger)^{1/2} (L_X^\dagger)^{1/2} P (L_Y^\dagger)^{1/2})^{1/2})$$

$$= \mathrm{tr}(((L_X^\dagger)^{1/2}P(L_Y^\dagger)^{1/2}(L_Y^\dagger)^{1/2}P^T(L_X^\dagger)^{1/2})^{1/2}) = \mathrm{tr}(((L_X^\dagger)^{1/2}PL_Y^\dagger P^T(L_X^\dagger)^{1/2})^{1/2})$$

so swapping $P$ and $P^T$, our formulas for the two distances are equal.

Finally, we check the triangle inequality. Let $X, Y,$ and $Z$ be graphs with $|X| = N, |Y| = M,$ and $|Z| = L$. Let $P$ and $T$ satisfy the conditions in the definition of $\Delta(X, Y)$ so that the integral is within $\epsilon$ of the minimum value. Similarly let $Q$ and $S$ satisfy the conditions in the definition of $\Delta(Y, Z)$. It suffices to show that

$$\left( \frac{1}{MN} \int_{\mathbb{R}^X} \sum_{x \in X} \sum_{y \in Y} (f(x) - (Tf)(y))^2 P(x, y) d\mu^X(f) \right)^{1/2} \tag{1}$$

$$+ \left( \frac{1}{NL} \int_{\mathbb{R}^Y} \sum_{y \in Y} \sum_{z \in Z} (f(y) - (Sf)(z))^2 Q(y, z) d\mu^Y(f) \right)^{1/2}$$

$$\geq \left( \frac{1}{ML} \int_{\mathbb{R}^X} \sum_{x \in X} \sum_{z \in Z} (f(x) - (S(Tf))(z))^2 \sum_{y \in Y} \frac{P(x, y)Q(y, z)}{N} d\mu^X(f) \right)^{1/2}.$$

because then $\sum_{y \in Y} \frac{P(x,y)Q(y,z)}{N}$ and $S \circ T$ satisfy the conditions in the definition of $\Delta(X, Z)$, and then taking $\epsilon$ sufficiently small, we obtain the triangle inequality.

This inequality follows from the Cauchy-Schwarz inequality

$$\left( \frac{1}{MNL} \int_{\mathbb{R}^X} \sum_{x \in X} \sum_{y \in Y} \sum_{z \in Z} (f(x) - (S(Tf))(z))^2 P(x, y)Q(y, z) d\mu^X(f) \right)^{1/2} \leq \tag{2}$$

$$\left( \frac{1}{MNL} \int_{\mathbb{R}^X} \sum_{x \in X} \sum_{y \in Y} \sum_{z \in Z} (f(x) - (Tf)(y))^2 P(x, y)Q(y, z) d\mu^X(f) \right)^{1/2} \tag{3}$$

$$+ \left( \frac{1}{MNL} \int_{\mathbb{R}^X} \sum_{x \in X} \sum_{y \in Y} \sum_{z \in Z} ((Tf)(y) - (S(Tf))(z))^2 P(x, y)Q(y, z) d\mu^X(f) \right)^{1/2} \tag{4}$$

where (3) simplifies to the first term of (1) by using $\sum_{z \in Z} Q(y, z) = L$ and (4) simplifies to the second term of (1) by using $\sum_{x \in X} P(x, y) = N$ and $T_\# \mu_X = \mu_Y$ so the integral against $\mu_X$ of a function of $Tf$ is equal to the integral against $\mu_Y$ of the same function of $f$.

$\square$

## 2 Further details on COPT optimization

Here we elaborate further on COPT's optimization routine. As the objective Equation 3.1 is not globally convex, gradient descent can fall into local minima. To facilitate faster convergence and convergence towards global minima, we use a combination of learning rate scheduling and learning rate hikes.

Specifically, the learning rate is initialized at $0.4$. During training, it's scaled multiplicatively by $0.7$ per 100 iterations. Once the *change* in loss drops below a threshold, set at $0.002$ throughout, for 10 different iterations, the learning rate is increased five-fold, capped at $4.0$. If an upper bound on the number of iterations is set, learning rate hikes stop 200 iterations before the max number of iterations, to allow convergence. These parameters can be tuned with respect to the downstream task, we have found COPT to be robust with respect to the parametrizations.

Figure 1 illustrates the effects learning rate hikes on training loss, where LR hiking is the only difference between the two curves. The loss increases briefly when the learning rate is hiked, but can soon drop below the level had there been no hikes. LR hiking not only improves convergence rate,

Figure 1: The impact of learning rate (LR) hiking on COPT training, in this case computing the alignment and distance between two 25-node graphs. The dashed line represents no LR hiking, the solid line represents LR hiking, the triangles indicate LR hikes. The hikes are triggered when the change in loss falls below $0.002$ over $10$ different iterations. Not only are the rates of convergence different, the presence or absence of LR hiking can lead to different permutations.

but experiments show it also changes the finaly transport plan $P$, indicating convergence to a better minimum.

We observe that this optimization routine allows for **faster runtime** per iteration than stochastic exploration used in GOT, which, to avoid converging to local minima, uses a stochastic exploration method that minimizes the *expectation* of the distance, rather than the distance itself. But this requires a nontrivial number (e.g. 10) of random explorations at each training step to achieve good performance.

For instance, when aligning the same set of $50$-node graphs, on the same machine and CPU, $1000$ iterations of GOT takes $18.8 \pm 0.72$ seconds, and COPT takes $3.18 \pm 0.59$ seconds, with settings that were tuned to produce the best community discovery results for each method, which as observed are commensurable (as described in §3). This is repeated 20 times.

## 3  Further comparison with GOT

Under the special case $N = M$, beacuse the transport plan $P$ converges to a permutation, COPT reduces to the GOT metric described in [4]. We prove this in Lemma 4.1. As this case uses the same optimization backbone when $N \neq M$ as well as for sketching, COPT and GOT differ in their implementations.

We want to compare the methods in the special case $N = M$, to ensure that, as in theory, the two implementations are commensurable.

However, the permutations between the two in the alignment task are not directly comparable, as many graphs carry symmetries that allow multiple transport plans to achieve the same objective minimum [1].

Thus, to compare our implementation with GOT's, we repeat one key alignment experiment in [4], where 40-node four-community graphs are aligned with corrupted and permuted versions of themselves, and the alignment quality is measured using the normalized mutual information (NMI).

Figure 2: Sketching a $50$-node graph down to $15$ nodes with COPT, which typically converges in $\sim 200$ iterations.

| #edge rem. | 30 | 60 | 90 | 120 | 150 | 180 | 210 | 240 |
|---|---|---|---|---|---|---|---|---|
| GOT | .997±.013 | 1.±0. | .991±.022 | .960±.059 | .837±.092 | .855±.10 | .844 ±.096 | .862±.11 |
| COPT | 1.±0. | 1.±0. | .994±.026 | .964±.094 | .873±.13 | .848±.13 | .897±.125 | .871±.14 |

Table 1: **Alignment quality as gauged by community discovery performance.** Normalized mutual information (NMI) scores between $40$-node graphs against corrupted and then permuted versions of themselves, using GOT and COPT implementations. Higher is better. First row indicates number of edges removed. The same upper bound on the number of iterations, 1000, is used for both methods, and parameters tuned. GOT is a special case of COPT when $N = M$, and the NMI above demonstrates that they are also commensurable empirically.

Specifically, we 1) randomly delete a given number of nodes, 2) permute nodes in the reduced graphs, 3) align between the original graphs and the permuted reduced graphs using COPT and GOT, and 4) compare the resulting NMI using the permutations produced by the alignment algorithms.

As shown in Table 1, the NMI are commensurable. Each measurement is repeated 20 times after parameter tuning. Furthermore, the GOT NMI are consistent with [4], which also report that these NMI scores outperform the NMI for the same evaluation settings using GW.

## 4 Characterization of minima

The following lemma characterizes the minima in the COPT objective function, in general and in the special case $N = M$.

**Lemma 4.1.** *Any local, hence global, minimum of the function*

$$\inf_{\substack{T:\mathbb{R}^X \to \mathbb{R}^Y \\ T_\# \mu^X = \mu^Y}} \int_{\mathbb{R}^X} \sum_{x \in X} \sum_{y \in Y} (f(x) - (Tf)(y))^2 P(x,y) d\mu^X(f)$$

*on the convex polyhedron*

$$\left\{ P : X \times Y \to \mathbb{R}^+ \mid \sum_{x \in X} P(x,y) = N, \sum_{y \in Y} P(x,y) = M \right\}$$

*that is isolated, in the sense that no other point within distance $\epsilon$ for any $\epsilon$ is a local minimum, is a vertex of that polyhedron.*

*In particular, if $N = M$, every isolated local minimum is a permutation matrix.*

Because local minima being isolated is a generic condition, this shows that the local minima are permutations away from a set of Laplacians $L_X, L_Y$ with measure $0$, and thus COPT will converge to a permutation outside a set of measure $0$.

*Proof.* Let $P_0$ be a local minimum of

$$\inf_{\substack{T:\mathbb{R}^X \to \mathbb{R}^Y \\ T_\# \mu^X = \mu^Y}} \int_{\mathbb{R}^X} \sum_{x \in X} \sum_{y \in Y} (f(x) - (Tf)(y))^2 P(x,y) d\mu^X(f) = \min_\gamma L(P,\gamma)$$

where the minimum is taken over measures $\gamma$ on $\mathbb{R}^X \times \mathbb{R}^Y$ whose projection to $\mathbb{R}^X$ is $\mu^X$ and whose projection to $\mathbb{R}^Y$ is $\mu^Y$, and where

$$L(P,\gamma) = \int_{\mathbb{R}^X} \int_{\mathbb{R}^y} \sum_{x \in X} \sum_{y \in Y} (f(x) - g(y))^2 P(x,y) d\gamma(f,g).$$

We switch from the transport map to the joint measure formulation of optimal transport as only for the joint measure is the minimum always attained.

Let $\gamma_0$ be the value of $\gamma$ which attains this minimum. Thus $P_0$ is also a local minimum of $L(P,\gamma_0)$. To prove this, note that if $L(P,\gamma_0)$ a smaller value at some nearby point, $\min_\gamma L(P,\gamma)$ a smaller

value, because $\gamma_0$ is included in the minimum over $\gamma$, and so $P_0$ will fail to be a local minimum of $\min_\gamma L(P, \gamma)$ assumed.

Because $L(P, \gamma_0)$ is a linear function of $P$, its local minima are simply faces of the polyhedron of possible values of $P$. If $P$ is not a vertex, then it lies in a positive-dimensional face, on which $L(P, \gamma_0)$ is constant, and thus on which every value of $\min_\gamma L(P, \gamma)$ is at most $L(P_0, \gamma_0) = \min_\gamma L(P, \gamma)$. In this situation, the only way $P_0$ can be a local minimum of $\min_\gamma L(P, \gamma)$ is if $\min_\gamma L(P, \gamma)$ is actually constant on a neighborhood of $P_0$ in this face. In that case, every point in the constant region is a local minimum, contradicting our assumption that $P$ is a local minimum.

Finally, we must check that every vertex if $N = M$ corresponds to a permutation. To see this, let $P$ be a vertex, and consider a bipartite graph with vertices $X \cup Y$ with an edge connecting $x \in X$ and $y \in Y$ if and only if $P(x, y) \neq 0$.

Let us check that this graph does not contain any cycle. If it did, because the graph is bipartite, we could 2-color each edge of the cycle. Then for any sufficiently small $\epsilon > 0$, raising $P(x, y)$ by $\epsilon$ for $(x, y)$ each red edge and lowering $P(x, y)$ by $\epsilon$ for $(x, y)$ each blue edge would preserve all the conditions on $P$ defining the polyhedron. Similarly, lowering $P(x, y)$ for the red edges and raising $P(x, y)$ for the blue edges would preserve the condition. Thus $P(x, y)$ is a convex combination of two different points in the polyhedron and hence is not a vertex.

Because this graph does not contain a cycle, all its connected components must be trees, and every tree contains a leaf. But if $x \in X$ is a leaf, then $P(x, y) > 0$ for a unique $y$, so $P(x, y) = M = N$ and thus $P(x', y) = 0$ for a unique $x'$. The same holds if $y \in Y$ is a leaf. Thus every leaf is connected to another leaf and so every tree is a single edge, so in fact every component of the graph is a single edge. Thus, each $x \in X$ is connected by exactly one edge to a unique $y \in Y$, and vice versa, defining a permutation $\sigma : X \to Y$ with $P(x, y) = \begin{cases} N & y = \sigma(x) \\ 0 & y \neq \sigma(x) \end{cases}$.

$\square$

## 5  Further details on datasets

Table 2 contains additional details on the graph datasets used in experiments, which recall are Proteins [1], BZR_MD [3], MSRC_9 [5], and Enzymes [6].

|          | # graphs | # classes | Avg # nodes | Avg # edges |
|----------|----------|-----------|-------------|-------------|
| Proteins | 1113     | 2         | 39.06       | 72.82       |
| BZR_MD   | 306      | 2         | 21.30       | 225.06      |
| MSRC_9   | 221      | 8         | 40.58       | 97.94       |
| Enzymes  | 600      | 6         | 32.63       | 62.14       |

Table 2: Details on experimental datasets.

Note that we work with connected, undirected graphs. The "connected" assumption is unnecessary, but we focus on connected graphs as they are more relevant for most applications. The "undirected" assumption is necessary because we define the variance-covariance matrix of $\mu^X$ as the pseudoinverse of the Laplacian. A variance-covariance matrix is always symmetric, so the Laplacian must be symmetric.

## 6  Further experiments

This section contains further experiments to show the effectiveness and utility of COPT for visualization and summarization tasks.

### 6.1  Graph projection to low dimensions

Projecting graphs to low dimensions can be an effective technique for visualizing relations amongst graphs, analogous to how projection techniques such as t-SNE [9] can yield important insights in the distribution of data points.

Figure 3: 3D sketches of 80 ten-node graphs (left) and zoomed in version (right) reveal that sketches of the same class cluster together or lie in the same subspace.

Here a set of $80$ ten-node graphs consisting evenly of four classes of graphs [2]: 2-block [2], random regular [7], powerlaw tree, and caveman [10], are sketched down to graphs on three nodes, and the three entries in the strict upper triangular part of the sketched Laplacian are *canonicalized* by sorting, and used as three-dimensional vector representations of the graphs.

As seen in Figure 3 and the zoomed in version, the 3D sketches of each class of graphs roughly follow the same trend, whether clustering together or lying in the same subspace.

## 6.2 Graph retrieval

Following up on §5.1, this subsection provides a more detailed comparison between the retrieval quality of different pipelines, namely retrieval using [COPT sketches + GW] and [spectral projections + GW], when both COPT sketching and spectral projections are allowed to reduce the original graphs to a *fixed* number of vertices. As seen in Figure 4, retrieval using COPT sketched Laplacians outperforms spectral projections of original Laplacians by a margin in accuracy, where accuracy is determined by whether the top-retrieved candidate has the same class as the query.

Figure 4: Comparing nearest-neighbor based classification between pipelines: [COPT sketches+GW] vs. [spectral projections+GW]. The x-axis indicates the number of candidates from the coarse method ($l^1$ retrieval on COPT sketches or $l^2$ retrieval on spectral projections) allowed to advance to the finer but more time-consuming method (GW). The query set and search dataset consist of 180 and 600 50-node graphs across six categories, respectively.

### 6.3 Graph summarization examples

We provide additional examples for graph summarization in Figure 9, in which graphs of varying initial structures are sketched down to a reduced number of nodes. As seen in these sketches, COPT is able to preserve the initial global structures of graphs, sketching structurally adjacent nodes in the original graph to the same node or nearby nodes in the reduced graph.

## Footnotes

[1]Inspection by hand reveals that, on "simple" graphs such as two-block graphs, the transport plans produced by COPT and GOT are the same up to symmetry.

[2]These classes were chosen as they admit meaningfully distinct properties on ten nodes, unlike for instance a four-community graph.

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

Figure 5: Lollipop.

Figure 6: Star.

Figure 7: Grid.

Figure 8: Ring.

Figure 9: **Graph summarization:** orginal graphs (above) and their sketched graphs (below). The node labels on the sketched graphs are determined using the transport plan $P$, specifically the label on a node contains the two top-weighted nodes in the original graph whose mass flowed into that node. COPT sketches structurally similar nodes in the original graph to the same or nearby nodes.