[Reviews · NeurIPS 2020]

Review 1

Summary and Contributions: The paper presents a novel scheme to compute a graph distance between two graphs (X, E) and (Y, E') by associating the vertex sets X and Y with an optimal transport plan and the graph signals at vertices x \in X and y \in Y with another (inner) optimal transport plan. The inner optimal transport can be optimized in closed form while the outer plan can be approximately optimized via a gradient descent approach. It also turns out that this setup yields a natural way to perform graph sketching, in particular to find a surrogate Laplacian of a graph that is still as close as possible to the original Laplacian in terms of the suggested distance, and that the surrogate Laplacian can be extended to a full surrogate graph via straightfoward thresholding techniques. The proposed graph sketching method is then evaluated experimentally in terms of graph classification and graph retrieval performance, while also adding some example instances of graph summarization.

Strengths: The paper's main strength is the derivation of its proposed method. The derivation starts with an intuitive but intractable measure of distance between graphs which is then approximated in a theoretically well-motivated fashion, yielding an efficicient and easy-to-understand computation approach. Additionally, the paper goes deep into how the method represents graphs and contrasts this with related work. This deep discussion then also yields a graph sketching approach which the evaluation focuses on. The entire derivation was a joy to read and seems exemplary to me. Further, optimal transport and graph processing have attracted considerable interest in the NeurIPS community in recent years, such that this paper should be relevant to a wide section of the community. It is also conceivable that some concepts introduced in this paper could be translated to graph neural network research (although the paper does not discuss this at the moment).

Weaknesses: The main weakness of the paper is, I believe, one of focus. The paper is set up as introducing a novel distance, but the main focus of the paper seems to be the graph sketching approach. There is no algorithmic description of computing the COPT graph distance, only of graph sketching, and the experiments do not evaluate the COPT distance at all but only the graph sketching approach. While the supplement does provide an additional alignment experiment (which I appreciate), the main paper falls short in this regard. This is particularly confusing to me because there are obvious ways in which the distance measure could have been integrated into the experiments. In particular, the graph classification could have been performed with a k-nearest neighbor classifier on the COPT graph distance, which could then have been compared to a k-nearest neighbor classifier on other distance measures. Further, the graph retrieval could also have been performed with the COPT distance instead of first vectorizing the graphs and then performing a nearest-neighbor search. As it stands, the experiments hardly provide any evaluation of COPT as an actual graph distance. I see two ways to address this point. Either, an algorithmic description for the COPT distance should be added, as well as further experiments (which I would recommend), or the paper could be limited in its scope and the abstract/introduction should be re-written to focus only on graph sketching and mention a graph distance measure only as a side-note. In addition to this main point, there are a few minor issues, most of which are discussed on the other sections below. One important point is regarding optimization: The paper proposes (projected) gradient descent to find the optimal transport plan, but it is not clear to me that this is optimal. Other optimization techniques, such as conjugate gradient, lbfgs, ADMM, etc. could be considered here, which might significantly speed up the approach.

Correctness: The theoretical arguments are, I believe, flawless and are provided both in detail and in a convincing high level summary in the main paper, both of which reflect a deep knowledge of the subject matter. The experimental methodology also appears solid to me, although room for improvement exists. In particular: * Instead of random repeats, a crossvalidation would be preferable, which would also enable statistical analysis. * The measure of variance in Table 1 is not described. I assume this is standard deviation, but this could be clarified. * It is not clear how often the synthetic experiments in 5.2 were repeated.

Clarity: Clarity is another important point that could be improved, I believe, which is mostly due to structure. * line 65: It would be more clear to say that d is a distance between vertices, which should take into accoung the edge structure. * In section 4 the structure coould be improved for clarity: The heading for section 4 is COPT for graph sketching, but the time complexity discussion in 4.2 seems to refer to both sketching and to the general graph distance problem. It would be more helpful, I believe, to have section 4 as subsection 3.3, subsection 4.1 as a paragraph/subsubsection, and subsection 4.2 as subsection 3.4. The related work section 4.3 should then be section 4, i.e. a separate section. * Discussions on related work are currently spread out over the paper, in particular section 2 can be seen as a related work section, and section 4.3 as well. I would recommend to merge these sections to a shared background/related work section, which would also leave more space for other points. * Regarding time complexity in 4.2, there seems to be some speedup potential that could be discussed. In particular, it would be helpful to discuss which parts of Equation (5) can be computed in advance and which parts need to be re-computed in each gradient step. I would also assume that the process is nicely parallelizeable, perhaps even on GPU. * The graph summarization approach in section 5.3 should be moved to the method section, I believe. It should also be discussed how the thresholds are set. /edit During discussion, further points of clarity emerged which should be addressed. In particular: * In equations (3-5) it seems like R^X and R^Y refers to the |X| and |Y|-dimensional real vector space, but as the authors clarified this instead refers to the spaces of functions from X to R and Y to R respectively. I understand that these are, in a sense, equivalent, because the function f from X to R can be written as a |X|-dimensional real vector (and for Y respectively). But it should nonetheless be clarified because it is otherwise hard to understand the meaning of the operator T. * To further clarify that T acts on f rather than the output of f it could be denoted as [T(f)](x) instead of T f(x). * In the proof to Lemma 3.1 it should be clarified how the matrix forms of A and B are obtained when A and B are operators on functions. Overall, I think, it is insufficiently clear that much of the arguments happen in the space of functions rather than the real vector space. I believe the arguments work, but it is a non-trivial cognitive jump for the reader and prevents deeper understanding of the approach when not clarified.

Relation to Prior Work: While the comparison to other work from the optimal transport domain is comprehensive and, I believe, complete, I would strongly recommend to also include graph edit distances in the discussion of related work. Such distances are strongly related (although not equivalent) to optimal transport and would also help to clarify some concepts. In particular, the computational hardness stated in lines 54-55 directly falls out of graph edit distance theory, because finding an optimal bijection between vertex sets would solve the graph edit distance problem, which is provably NP-hard (see e.g. Bougleux et al., 2017, doi: 10.1016/j.patrec.2016.10.001 ).

Reproducibility: Yes

Additional Feedback: With regards to the broader impact statement, the paper only suggests positive impacts and does not refer to ethical hazards. Although I would be hard-pressed, as well, to find dangeorus applications of this method, I would still recommend to caution against over-interpreting the graph distance measure: if this method regards two graphs is similar, this makes no statement about graph 'semantics' but only about structure. Regarding reproducibility, I do not think that the README is sufficient to reproduce the results in the paper without additional effort (there is no description of which steps should be run, exactly, to reproduce the experiments). However, the details given in the paper in conjunction with the code documentation should suffice to make this effort manageable. I would like to add that, despite the many shortcomings mentioned, I regard the main contribution of the paper, namely the novel graph distance and graph sketching aproach, as outstanding. This outweighs, in my mind, the shortcomings, such that I would regard the paper still as a clear accept, with potential for stellar quality if further shortcomings are addressed.


Review 2

Summary and Contributions: This paper introduces a new metric between graphs based on OT, that takes into account spectral information. The paper is primarily empirical: (i) although computing this distance is a non-convex optimization problem, they show empirically that gradient descent converges fast to a stationary point; (ii) they provide impressive, detailed empirical results comparing their algorithm to many existing algorithms on a range of tasks (sketching, retrieval, classification, and summarization).

Strengths: The problem of graph sketching is an important and well-motivated problem. As mentioned in the paper, the desiderata for what type of graph sketch is desired depends on the application. The new graph metric defined in this paper gives rise to graph sketching algorithm that is empirically shown to outperform existing approaches from the ML community in several applications. The experimental results are comprehensive, and clearly significant effort has gone into doing a thorough comparison with existing approaches from the ML community in several applications of practical interest.

Weaknesses: While the paper thoroughly describes graph sketching algorithms from the ML community, it appears to entirely miss a large body of literature about Laplacian sketching from the optimization and theoretical computer science communities. See eg the survey "Twice-Ramanujan Sparsifiers" by Batson-Spielman-Srivastava, SIAM Review 2014, and the papers before/after it, eg "On Sketching Quadratic Forms", "Efficient O(n/eps) Spectral Sketches for the Laplacian and its Pseudoinverse", etc. This line of work develops sparse graph sketches that (i) provably preserve the Laplacians spectrum up to arbitrary error, and (ii) provably run in near-linear time. It seems that the current paper can provide neither such guarantees (as the spectral properties of the optimum of the proposed work seem hard to analyze, and also the non-convexity of the problem means that GD can only find stationary points, not minimizers)? But perhaps the proposed method is better empirically? It would be good to investigate the tradeoffs. The fact that the proposed metric is a nonconvex optimization and thus can only be solved by heuristics is not mentioned until page 6 (and even there it is not highlighted). This drawback should be mentioned much earlier in the paper, arguably even in the abstract. Section 3.2 nicely discusses some differences between spectral and metric approximations of graphs. But it would be helpful to make more explicit why one *wants* one or the other in downstream applications. Can the whole paper be generalized to weighted graphs? If so, it seems to me that this would help not just for encompassing more applications, but also to avoid the issue described in Section 4 about having to relax the sketches to obtain weighted graphs.

Correctness: Yes.

Clarity: Yes. a few typos - line 22, structured - line 61, R cup {0} - line 88, Wasserstein

Relation to Prior Work: The prior work seems thorough about approaches for the ML community, but misses a large body of highly relevant work from the optimization/TCS communities (see above).

Reproducibility: Yes

Additional Feedback: I think this paper has some nice ideas. My current score primarily reflects the missing relations to prior work. Minor comments: - Table 1: it'd be helpful to report some timing information, at least enough to give the reader a rough sense of the relative timings of the compared approaches. - line 150: The following sentence is imprecise enough to be easily argued against. Consider rephrasing. "This is a discrete optimization problem, and continuous optimization problems are computationally simpler." - Section 2, first paragraph. Graph isomorphism is a difficult computational problem with a long line of work. Consider adding a reference to point the readers to it. - Section 3.2, eqn. This formula is classical, consider adding a reference. - Section 4.2: Consider adding a remark clarifying that the reported fast matrix multiplication times are not actually used in practice (I'd assume). - The new metric is not defined until pg 3. Consider moving earlier so that the reader can get to your contribution more directly. ==After rebuttal: Thank you for addressing my concerns. Based on these updates, I have adjusted my score.


Review 3

Summary and Contributions: The authors propose an extension of Graph Optimal Transport (GOT) for the optimal transport between graphs having different number of vertices by introducing the transportation plan (Kantorovich formulation) instead of the permutation map (Monge formulation). The authors name it as Coordinated Optimal Transport (COPT). The authors show COPT preserves important global structural information on graphs (spectral information). Empirically, the authors show the advantage of COPT for graph sketching, graph retrieval and graph summarization.

Strengths: + The authors extend GOT for optimal transport between graphs having different number of vertices. + The proposed COPT performs well for graph sketching, graph retrieval and graph summarization.

Weaknesses: My main concern is as follow: + The smooth graph signals representation is a Gaussian with mean zero and variance-covariance matrix – inverse of Laplacian matrix. For graphs having different number of vertices, the Gaussian representations have different dimensions. It is unclear how the authors extend the optimal transport between two Gaussian distributions having the same dimension as in GOT into “optimal transport” between two Gaussian distributions having different dimension?

Correctness: I have some following concerns: + It is unclear how the authors extend the optimal transport between Gaussian distributions having the same dimension as in GOT into “optimal transport” between Gaussian distributions having different dimensions. (since optimal transport is also defined for probability measures having the same dimension). I have not carefully checked the proof for Lemma 3.1. yet, but it seems the authors map R^X, R^Y into a common space R^{X x Y} to compute the optimal transport. However, it seems that it is not the case for its definition. It is better in case the authors give more detail discussion how the authors overcome the different dimensions of Gaussian distributions of the graph signal representation. The current representation is not clear yet. What is the connection between the proposed COPT with the reviewed Gromov-Wasserstein distance in Equ. (2)? + The theoretical analysis about global reservation in section 3.2. is interesting, but seems superficial. The definition of COPT is based on the graph spectral representation of graphs (Gaussian distribution with zero mean and inverse Laplacian matrix as its covariance), then the OT between those graph spectral representations. While the definition of GW is based on pair-wise distance between supports, and then quadratic optimal transport for those pair-wise supports. It seems that the analysis in Section 3.2. may not clearly show that COPT reserve the global structure preservation (spectral information) as claimed yet.

Clarity: The paper is easy to follow. However, it seems better in case the authors give more details for the proposed COPT, and the theoretical analysis for the global structure reservation of COPT.

Relation to Prior Work: Yes

Reproducibility: Yes

Additional Feedback: After the rebuttal: I thank the authors for your rebuttal. In my opinion, the main contribution which is the coordinated optimal transport between 2 graphs with different nodes may not clear enough yet. Therefore, I keep my current score. Even with further information from the discussions of area chair and authors together with the rebuttal, the formulation of COPT is not clear enough yet. The authors use graph signal to represent a graph as a Gaussian distribution (zero mean, inverse of Laplacian as its covariance). Therefore, the problem to compare 2 graphs with different number of nodes can be casted as a comparison between two Gaussian distributions in different dimensional spaces. (while standard OT can be only applied for probability measures in the same space, e.g., for the case of Gaussian distributions in the same space as in the GOT paper). The authors should emphasize "their special tool" to address this problem (which I guess the integration over the d\mu(f) instead of integration over supports of \mu with d\mu(x)) + What is the meaning of integration over d\mu(f)? If it is equivalent to integration d\mu(x), please proof. Otherwise, please compare them (since the authors also use the optimal transport between two Gaussian distributions in the same space in their proof, but there is no change form imo from integration over d\mu(f) to integration over d\mu(x)? + The meaning of f seems very important in the proposed formula (somehow, it can address the probability measure in different spaces with OT style?). However, the current information in the manuscript about f seems very limited (if not, almost no information about f). + it seems that the authors use T as the transformation between T: XX --> YY where XX is a space of functions from R^X --> R, and similar for YY as a space of functions from R^Y --> R (as in their additional information for area chair, without mentioned in manuscript). I am curious about its discrete form. I am also not clear why it is "equivalent/related" to standard OT. It is better in case the authors can address the clarity of the main contribution. ================== + In Equation (3) and (4), it seems that the authors have not explained the notation f? See more in the above Weakness, Correctness, Clarity Sections


Review 4

Summary and Contributions: The authors present a novel distance metric between graphs that minimizes a loss function over transport plans between two distributions: - The vertices of the graphs - The Laplacian spectra This novel distance metric seems to yield (based on the authors' experiments) insightful coarsenings of the graphs that can then be used to do classification, visualization, etc.

Strengths: The problem of correctly coarsening graphs or performing graph sketches is an extremely interesting one --- this could be used in particular over a wide variety of applications (brain connectomics, protein design, classification,etc.) The paper proposes an interesting extension of the GOT distance between graph that allows it to compare graphs of different sizes. The experiments that have been led on 4 independent datasets for graph classification are convincing (the presence of the interquantile range is especially appreciable). The additional experiment for graph summarization is also quite convincing.

Weaknesses: Maybe, as a way to further enhance the paper, it could have been interesting to compare the distance with other graph distances (in particular, the WL kernel). Here most of the experiments assess its performance as a way to perform graph coarsening, but a comparison with the WL kernel as a way to distinguish different graphs could have been insightful too.

Correctness: The method described and the claims seem to hold, to the best of my assessment.

Clarity: The paper is well written and the examples and experiments insightful.

Relation to Prior Work: The relationship is clearly discussed. Maybe a comparison with the WL kernel as a way to distinguish different graphs could have been insightful too (see weaknesses above).

Reproducibility: Yes

Additional Feedback:

[Author Response · NeurIPS 2020]

**R1** We appreciate your very constructive comments. Good point. Some new experiments comparing COPT distance to SOTA GW distance: With 6 classes of synthetic graphs, 18 queries on 60 graphs, with nearest-neighbors-based classifier, COPT achieves acc. 83.5±3.4, and GW 62.2±6.6. This is repeated 20 times. Will expand and include these. (The existing vectorize-and-retrieve experiments were designed to be highly practical on GPUs, leveraging the benefits of COPT during the vectorization preprocessing.) • Will update paper to emphasize distance aspect as you suggest. Algorithm#1 indeed already computes the COPT distance as the objective during optimization. • Indeed interesting to vary optimizer. To sketch a 20-node graph down to 5, currently each iteration takes 3.7±.05 ms, after updating to LBFGS, each iteration takes 128±3 ms. Likely because LBFGS is a second order method, and may evaluate the forward pass multiple times per iteration. Will try others. • There are in fact interesting relations to graph edit distance: in COPT, the matrix $P$ allows *continuous* (e.g. partial) flow between vertices, while edit distance involves matching the vertices *one-to-one* (plus insertions & deletions). Edit distance gives the *same cost* to delete an arbitrary edge, while COPT distance is higher if the edges deleted were important to the graph spectrum (i.e. edges connecting distinct clusters). A continuous relaxation of edit distance would resemble [LFF+15] more than COPT for this reason. • Added the fact the measure of variance is standard deviation, number of times 5.2 is repeated, will update broader impact & add details to README. Thanks for these. [LFF+15: Lyzinski et al, Graph matching: Relax at your own risk. 2015.]

**R2** We thank you for your points. Indeed Batson-Spielman-Srivastava and related works from TCS are fascinating. The primary reason for choosing the included literature was an alignment in goals. COPT was designed to reduce the number of *vertices*, producing *dense* Laplacians; BSS and related works reduce the number of *edges*, producing *sparse* Laplacians. One benefit of dense data is that they are well-suited for efficient processing on GPUs, such as in many ML applications. • BSS focuses on proving theoretical guarantees on constructing graph sparsifiers, it uses a *discrete* optimization method where one edge is added to the sparse graph per step (though weights are chosen continuously); COPT uses a *continuous* optimization method where every variable can change at every time step. Though discrete optimization can more readily produce provable guarantees, continuous optimization is often preferred in practice, e.g. continuous relaxations of discrete problems such as LP relaxation. • Another effect of the proof-driven construction is that the constructed sparsifier may not be useful for applications such as graph classification. E.g. BSS sparsifies complete graphs to sparse expander graphs, which have intricate structures not present in original graph. Attempts to classify the sparsifier could see this structure instead of properties of the original graph. COPT empirically reduces complete graphs to complete or nearly-complete graphs. • We will add these. • Similarly, the approach (e.g. in Jambulapati-Sidford) of defining *multiple* randomized sketches and then taking the median to obtain a good Laplacian approximation may not be practical for applications. • Good call on motivating §3.2 with downstream applications, *metric* approximations are good for applications where shortest paths between nodes are important, e.g. classifying road networks, classifying graphs arising from physical objects where edge lengths carry geometric information. *Spectral* approximations are useful for when the number of paths matter more, e.g. graph partitioning, where one aims to minimize the number of edges cut (one way is to iteratively coarsen graphs and find cuts on smaller graphs). Similarly COPT is applicable to graph clustering. • Will state nonconvexity earlier in the paper; comment on the theoretical fast matrix multiplication methods vs practice (indeed the theoretical fastest method is not practical); add references as suggested; comment on runtimes; and fix typos. • The only change to work with weighted graphs is to use the *weighted* Laplacian instead of ordinary Laplacian. The proofs only use the fact that $L_X$ is a symmetric PSD matrix. We brought in the additional concept of weightings only when needed, but the potential for applications to weighted graphs is a good point.

**R3** In Eq. (4), we define a loss function for a transport map between $\mathbb{R}^X$ and $\mathbb{R}^Y$, as a generalization of the loss in GOT from permutations to arbitrary matrices $P$. A priori this is just a formula, but in Lemma 3.1, we prove it is equal to a distance in $\mathbb{R}^X \times \mathbb{R}^Y$, after embedding $\mathbb{R}^X$ and $\mathbb{R}^Y$ into $\mathbb{R}^{X \times Y}$, which we use to calculate it. In the special case when $P$ is a permutation, these embeddings have the same image, reducing the formula to optimal transport in just $\mathbb{R}^X$ or $\mathbb{R}^Y$ as in GOT. • §3.2 compares COPT with GW, and discusses how COPT preserves paths-counting in relation to the spectrum. • $f$ is a free variable in the domain of integration, i.e. the space of functions on $\mathbb{R}$. Will specify this in the paper. Thanks.

**R4** Thanks much for your comments. It is indeed useful to compare with the WL kernel. Some new experiments: On 60 synthetic graphs across 6 categories with 18 query graphs, using exact nearest-neighbor search with COPT as a distance: acc. 83.7±3.9; SVM with WL kernel: 80.5±9.0. This is repeated 20 times. • Choosing between the two depends on the task and the dataset: For a large dataset of size $N$, we expect a WL kernel-based SVM to be more practical than using linear scan with COPT distance. As an SVM can be trained during preprocessing, allowing *constant* time query classification, whereas exact nearest-neighbor search requires time *linear* in $N$. • For small datasets, COPT distance can be advantageous: with only three samples per class, using nearest-neighbor classification with COPT distance gives accuracy 90.7±5.2, and the WL kernel-based SVM gives 77.8±7.1. • For tasks such as predicting *continuous* attributes, an SVM with WL kernel can't readily be applied, but a nearest-neighbors classifier with COPT distance still can. • Will expand on these and add to the paper.

[Meta-Review · NeurIPS 2020]

The paper introduces a new metric between graphs based on OT, that takes into account spectral information. The contribution has been deemed novel, strong and relevant by the reviewers with a comprehensive empirical validation. However, some part of the works lack a bit of clarity and we urge the authors to improve on these points for the final version [see comments of Rev 1 and 4].